# The Impact of Sport Activity Shut down during the COVID-19 Pandemic on Children, Adolescents, and Young Adults: Was It Worthwhile?

**DOI:** 10.3390/ijerph19137908

**Published:** 2022-06-28

**Authors:** Sara Raimondi, Giulio Cammarata, Giovanna Testa, Federica Bellerba, Federica Galli, Patrizia Gnagnarella, Maria Luisa Iannuzzo, Dorotea Ricci, Alessandro Sartorio, Clementina Sasso, Gabriella Pravettoni, Sara Gandini

**Affiliations:** 1Molecular and Pharmaco-Epidemiology Unit, Department of Experimental Oncology, IEO, European Institute of Oncology IRCCS, 20139 Milan, Italy; giulio.cammarata@gmail.com (G.C.); federica.bellerba@ieo.it (F.B.); sara.gandini@ieo.it (S.G.); 2Independent Researcher, 20121 Milan, Italy; giovanna.testa@fastwebnet.it; 3Department of Dynamic and Clinical Psychology, and Health Studies, Faculty of Medicine and Psychology, Sapienza University of Rome, 00185 Rome, Italy; f.galli@uniroma1.it; 4Division of Epidemiology and Biostatistics, IEO, European Institute of Oncology IRCCS, 20141 Milan, Italy; patrizia.gnagnarella@ieo.it; 5AULSS 9 Scaligera-Dipartimento di Prevenzione-UOC Medicina Legale, 37139 Verona, Italy; marialuisa.iannuzzo@gmail.com; 6Department of Human Sciences, MEDIS-Maxi Emergencies International School, 47890 San Marino, San Marino; doroteari@libero.it; 7Experimental Laboratory for Auxo-Endocrinological Research, Istituto Auxologico Italiano, IRCCS, 28824 Piancavallo, Italy; sartorio@auxologico.it or; 8Division of Auxology and Metabolic Diseases, Istituto Auxologico Italiano, IRCCS, 28824 Piancavallo, Italy; 9Crescere Sani Onlus, 20121 Milan, Italy; 10INAF-Osservatorio Astronomico di Capodimonte, 80131 Naples, Italy; clementina.sasso@inaf.it; 11Applied Research Division for Cognitive and Psychological Science, IEO, European Institute of Oncology IRCCS, 20141 Milan, Italy; gabriella.pravettoni@ieo.it

**Keywords:** COVID-19, children, adolescents, young adults, sport closure, lockdown

## Abstract

We conducted a national retrospective survey of 1764 athletes aged ≤25 years to investigate the benefit–risk balance of sport closure during the COVID-19 pandemic peaks in Italy. Univariate and multivariable analyses were carried out to investigate the association between sport practice during the study period and (1) the risk of SARS-CoV-2 infection in athletes and their families and (2) body mass index (BMI) change, and adherence to World Health Organization (WHO) guidelines for physical activity. The percentage of subjects with a positive SARS-CoV-2 test was similar in those participating and not taking part into sport activities (11% vs. 12%, respectively, *p* = 0.31). Restricting the analysis to subjects who practiced sports within an organized sport society/center, the risk of SARS-CoV-2 positivity was reduced for athletes who had never stopped their training (odds ratio (OR); 95% confidence intervals (CI): 0.62; 0.41–0.93). On the other side, responders who had stopped sport activity showed a 1% increase in BMI. Adherence to WHO guidelines for physical activity was significantly higher for athletes who had continued sport activities. In conclusion, sport closure and limitations had an important negative impact on the overall health of young athletes, being also not effective in reducing the spread of COVID-19.

## 1. Introduction

The coronavirus disease 2019 (COVID-19) was first identified in China in December 2019 [1], and was declared a pandemic by the World Health Organization (WHO) in March 2020. Many governments decided to introduce interventions to diminish the spread of the virus, including lockdowns [2], which forced many people to stay at home and limit their physical activity (PA) [3], in turn increasing the incidence of obesity and related health risks [4]. During the autumn of 2020, a second wave was registered in Italy and in most European and North American countries, leading to new limitations that determined a sport deprivation for most people. More recently, participation in sport activities was prohibited in Italy to all the non-vaccinated athletes older than 12 years, thus forcing many adolescents and young adults to physical inactivity.

It is well known that transmission of the SARS-CoV-2 is primarily related to direct exposure to respiratory droplets [5,6] and it was suggested that their spread could be potentiated by high-intensity exercise [6], thus leading to the decision of sport shut down as a preventive measure. However, very few epidemiological data were available on the causal association between practicing a sport and increased risk of infection, with contrasting results [6,7,8,9,10,11,12,13,14].

On the other hand, it is well known that sports play an important role in the physical and mental wellbeing of children and adolescents [15]. A higher amount of PA during childhood has multiple beneficial effects for cardiorespiratory and muscular fitness, cardiometabolic health, and bone health and is key in preventing heart diseases, type 2 diabetes, and cancer [16]. Despite the WHO recommendation, previous studies show that most children and adolescents had an insufficient level of PA [17,18], so much so that physical inactivity was defined as a pandemic issue [19]. The “inactivity pandemic” has potential detrimental consequences on physical wellbeing, especially in children with chronic diseases, disabilities, and special health needs [20,21,22]. Moreover, risk factors for severe COVID-19 disease include heart or lung problems, diabetes, or obesity, and these risk factors are closely linked to sedentary lifestyles and physical inactivity. Experts [23,24] suggested that the COVID-19 preventive strategies may lead to a decrease in PA and a worsening of depressive symptoms particularly in areas of poverty, which are already highly affected by COVID-19 morbidity and mortality.

The purpose of our study was to investigate the benefit–risk balance of sport closure during the COVID-19 pandemic peaks in Italy. Specifically, the first aim of the study was to assess the association of sport activity shut down with the risk of SARS-CoV-2 infection in athletes in order to understand whether this preventive measure was actually effective in reducing SARS-CoV-2 spread in young athletes and their families. The second aim was to evaluate the impact of sport activity shut down on the physical health of children and adolescents, with focus on weight increase and compliance with WHO guidelines for PA.

## 2. Materials and Methods

Between June and September 2021, a national retrospective cross-sectional survey of children, adolescents and young adults (aged ≤ 25 years) who used to play sports was performed. The survey was conducted using the Google Forms web survey platform, and the link was shared throughout sport clubs and social media. Responders older than 25 years and professional athletes were not allowed to proceed with the questionnaire based on our exclusion criteria. The questionnaire was developed by a panel of experts with in-depth knowledge in different areas, among which were epidemiology, nutrition, and psychology. It was in Italian and required an estimated time of 12–15 min to be completed. Survey administration was anticipated by a two-week pilot phase in which we tested the reliability of the adopted questionnaire and clarity of the questions. The original version of the questionnaire is provided (Appendix A). The questionnaire included multiple choice and open-ended questions divided into six different sections: sport-related questions, SARS-CoV-2 infection, socio-demographic, PA and mental health, diet, and screen time. Participants were first asked whether they had practiced a sport activity in the period between September 2020 and May 2021. If yes, they answered to several sport-related questions, including: sport type, number and two-week period of training sessions, participation in sport competitions, participation in sport activities organized by sport societies or sport centers, individual/team training, indoor/outdoor activities, and preventive measures (multiple choice among none, distancing, face masks during training, environment disinfection, aeration, use of lockers and showers, triage, other to be specified). The section of SARS-CoV-2 infection included questions on the number and date of positive test(s) for athletes and their cohabitants, as well as the total number of SARS-CoV-2 tests for athletes. Socio-demographic questions included gender, age, geographical area, height and weight in September 2020 and May 2021, school attendance, number of cohabitants, parents’ education, parents’ worksite, presence of outdoor spaces at home, and outdoor physical activity. The physical activity and mental health section included an adaptation of the International Physical Activity Questionnaire (IPAQ) in which physical activity was referred to the within the period of COVID-19 picks instead of the last week. Food consumption was investigated using a 16-item validated questionnaire that measures the daily or weekly intake of the main food groups and adherence to the Mediterranean diet in the study period. Finally, we asked for the screen time, including television, personal computer, tablet, smartphone, and videogames. An extended version of Materials and Methods and the original version of the questionnaire are provided in the Appendix A and Appendix A.

The funding source has no role in the collection, analysis, and interpretation of the data; in the writing the report, and in the decision to submit the paper for publication.

We collected data from 2910 questionnaires. We excluded 790 questionnaires according to exclusion criteria (N = 540 age > 25 years; N = 245 professional athletes) or for the lack of informed consent to participate (N = 5). From the remaining 2120 questionnaires, we removed 356 questionnaires after quality checks, thus the remaining 1764 questionnaires were to be used for the analysis (Appendix A).

### Statistical Analysis

In order to assess the association between sport activity and SARS-CoV-2 positivity during the study period, we defined two variables: “training by closure period”, which refers to sport activity practice during the closure period (November 2020–March 2021), only in opening periods (September/October 2020 and April/May 2021), or none; and “weekly training sessions”, which refers to the average number of weekly trainings.

The chi-square test and the non-parametric Wilcoxon-sum rank test were performed to investigate the association between any SARS-CoV-2 positivity and the study characteristics of the responders for categorical and continues variables, respectively. At multivariable analysis, an odds ratio (OR) with 95% confidence interval (CI) was calculated for the risk of being tested positive for SARS-CoV-2 at least once during the study period with logistic regression models. In order to investigate the robustness of the results, a generalized linear model analysis for the time point of SARS-CoV-2 positivity with repeated measures was also performed. All the main analyses were reported also for at least one SARS-CoV-2 positivity in the family and for the subgroup of athletes who declared that they had practiced sport activities organized by a sport society/center.

Among the athletes who participated in any sport activity, we investigated the association between preventive measures and other sport-related characteristics with the risk of any SARS-CoV-2 positive test with logistic regression analysis.

The association between responders’ characteristics with logarithm of body mass index (BMI) change (September 2020–May 2021) was assessed at univariate analysis with the Wilcoxon sum rank test for categorical variables and the Spearman correlation for continuous variables. As a multivariable analysis we used a generalized linear model with a logarithm of BMI change as a dependent variable and adjustment for the logarithm of baseline BMI and relevant covariates. The analysis was stratified for normal weight and overweight patients, defined according to age [25]. Adherence to the WHO guidelines, as reported in [26], was defined based on athletes’ age.

*p*-Values < 0.05 were considered statistically significant. The analyses were performed with R version 4.1.1 and SAS version 9.4 (SAS Inst., Cary, NC, USA) software.

## 3. Results

### 3.1. Positivity to SARS-CoV-2 Test

Baseline characteristics of the subjects are reported in Table 1 and Appendix A.

The median age (interquartile range (QR)) of responders was 12 (9–15) years, with a gender balance between males (55%) and females (45%). The majority of athletes were children in elementary school (41%), while a lower number of responders attended university (9%). The majority of responders (N = 1258, 72%) declared they had participated in at least one training session, while 492 (28%) stopped their sport activity.

Percentages of responders with a positive SARS-CoV-2 test in the study period were similar between those participating and not participating in sport activities (11% vs. 12%, respectively, *p* = 0.31, Table 1). Similar percentages of SARS-CoV-2 positive subjects were also reported for athletes who joined sport activities in different closure periods (*p* = 0.55, Table 1) and according to the number of training sessions (*p* = 0.10, Table 1). These results were confirmed from multivariable analysis (Table 2 and Appendix A), also taking into account repeated measures analysis (results not shown).

Factors associated with SARS-CoV-2 positivity at multivariable analysis were male gender, number of tests, attending professional high school and over compared to elementary school, low parents’ education, and parents’ working at least partially outside (Table 2). Interestingly, when the analysis was restricted to subjects who declared they practiced sport activity within an organized sport society/center, the risk of SARS-CoV-2 positivity was 38% lower for athletes who had never stopped their training (OR; 95% CI: 0.62; 0.41–0.93), compared to athletes who had stopped their sport activities. In the repeated measures analyses, taking into account the direct link between SARS-CoV-2 positive test and participation in sport activities, a similar trend, although not significant (OR; 95% CI: 0.83; 0.58–1.19, results not shown), was observed.

In order to account for the possible transmission of SARS-CoV-2 infection from asymptomatic young athletes to their families, the presence of at least one SARS-CoV-2 positive test among athletes’ cohabitants was investigated. The results are reported in Table 3 according to training by closure periods and in Appendix A according to the number of training sessions.

The probability of having a positive SARS-CoV-2 test in the family was similar for responders who practiced sports in the opening (OR; 95% CI: 1.20; 0.80–1.80) and closure (0.98; 0.71–1.35) periods compared to those who had stopped sport activity. Otherwise, an increased risk of SARS-CoV-2 infection in the family was suggested to be related to a lower parents’ education level and to parents’ working outside. Moreover, cohabitants of young athletes attending professional high school and over had a higher probability of being infected compared to those of children from elementary school (Table 3). A 20% increase of being tested positive for COVID-19 SARS-CoV-2 was observed when in the number of tests and of cohabitants was increased. Finally, a lower percentage of SARS-CoV-2 positivity was detected in central Italian regions compared to the Northern part (Table 3). Similar results were observed for the analysis stratified to athletes attending sport centers, with a 22% reduced risk (although not significant) of SARS-CoV-2 infection in the families of athletes who had continued to practice sports (OR; 95% CI: 0.78; 0.56–1.08).

### 3.2. Preventive Measures

After adjusting for athletes’ characteristics and sport-related covariates, we found that the most effective preventive measure reported in the questionnaire was to perform a triage (i.e., measuring fever and account for other symptoms and contact tracing), which reduced the risk of being infected by approximately 64% (OR; 95% CI: 0.36; 0.22–0.59, Figure 1). The other effective measures were avoiding the use of lockers and showers, which reduced the risk of being infected by approximately 45% (OR; 95% CI: 0.55; 0.33–0.91). Distancing and aeration were also associated with a significant risk reduction in multivariable analysis (Appendix A). The percentage of reported SARS-CoV-2 positivity was similar among different sports (*p* = 0.17, Appendix A).

### 3.3. BMI Change and Adherence to WHO Guidelines for PA

Figure 2 shows the percent change of BMI from September 2020 to May 2021 as reported in the questionnaires. Responders who completely stopped sport activity reported a 1% increase of BMI, while for athletes who participated to sport activities, we observed an apparent dose response effect (0% for those attending training sessions only in the opening period and −1% for closure periods) (Figure 2a). A similar trend was observed also according to the number of weekly training sessions (Appendix A). The benefit of sport activity seemed evident both for overweight and normal weight athletes. Specifically, overweight athletes reported an almost stable BMI if they stopped sports, while they reported a BMI decrease with participation in sport activities (Figure 2b and Appendix A); on the other hand, normal-weight athletes reported a BMI increase (+2% and +1%, respectively) if they stopped sports and a stable BMI with participation in sport activities (Figure 2c and Appendix A). Multivariable analysis confirmed these results (Appendix A). Other factors associated with BMI increase were: female gender, older ages (high school and over), South/Island location, and screen time (Appendix A). Factors associated with a reduced BMI were adherence to WHO guidelines and outdoor PA (Appendix A).

Overall, only 34% of responders reported levels of PA compliant with the WHO guidelines (Table 1). Notably, physical inactivity was significantly higher both in the group of athletes practicing sports only during opening periods (OR; 95% CI: 1.51; 1.09–2.10) and those who completely stopped sport practice (2.15; 1.54–3.02) compared to athletes who continued the sport activities (Appendix A). Other factors associated with increased physical inactivity were: older ages, with children of the elementary school more active that the older ones; practicing contact sports; no participation in matches or competitions; no outdoor space at home; no outdoor PA; daily screen time; and non-adherence to a Mediterranean diet.

## 4. Discussion

This is the first national study that investigated the impact of sport closure on SARS-CoV-2 infection and the overall health of young athletes. Our data show that sport closure and limitations were not effective: the percentage of responders with a positive SARS-CoV-2 test in the study period was indeed similar when comparing athletes who stopped their sport activity in the study period with those who continued their training sessions. Surprisingly, we observed an even significantly lower probability of being SARS-CoV-2 positive for subjects practicing sports in organized sport centers/societies compared to those not practicing sports at all. This result may be partially explained by an improved immune system as a result of sport activity that protected healthy athletes from severe COVID-19 infection. Indeed, it was suggested that exercise training increases leukocytes, and reduces inflammatory cytokines and chemokines [27,28,29], and that regular PA can reduce the risk of contracting a virus by 31% [30]. Moreover, because sociality is especially important for young subjects, it seems plausible to hypothesize that responders not practicing sports frequented other social activities in a less protected environment than sport centers, where specific preventive protocols were developed and applied.

Notably, our results refer to a population of largely unvaccinated children and young adults because the study period anticipated the advent of COVID-19 vaccination in Italy, which started for healthy young people only in June 2021. This should be remarkable also in the context of recent Italian policies that prohibited sport participation to all the unvaccinated athletes older than 12 years old.

Previous case reports have alarmed on the possibility of clusters within sport teams [7,8,9], leading the Centers for Disease Control and Prevention to recommend caution [10]. By contrast, recent published studies reported that sport itself cannot be considered a risky activity [6,11,12,13,14]. Even in contact sports such as American football, the number and duration of contacts were found to be very low and inconsistent with the hypothesis of a probable transmission [31].

A further reason for sport activity shut down was that COVID-19 was hypothesized to spread from young athletes (usually at low risk of severe disease) to frail co-habitants in their families, although this hypothesis was never before tested. Our results suggest that no increased risk of SARS-CoV-2 infection in the family could be attributed to the participation of young athletes in sport activities.

Among the preventive measures used in sport centers, we found that the triage and the prohibition on using lockers and showers were the most effective. Previous studies demonstrated that the symptomatic period is the one with the highest probability of infection, thus suggesting to avoid practicing sports with fever or other important symptoms in order to effectively prevent SARS-CoV-2 spread. A previous case report of a hockey team [8] confirmed that the closure of locker rooms and limited access to the building were the most effective measures. On the other side, the use of facemasks during sport activity resulted in having no impact on the SARS-CoV-2 spread. Similar results were reported in another study [6] and may be explained by the fact that facemasks used during sport activities become wet or a choking hazard, possibly increasing the virus proliferation. Additionally for this reason, the American Academy of Pediatrics [32] advised against the use of facemasks during sport activities.

Looking at athletes’ physical health, we found that subjects who stopped sport activity had a significant increase of BMI compared with those who continued to practicing sports. Interestingly, in the subgroup of overweight subjects, sport seemed to be effective in reducing BMI, while in the subgroup of normal weight subjects, it was helpful to maintain a BMI within the normal range. It was previously noted that the lockdown caused obesity in adolescents [33,34,35,36] and exposed those with obesity to a greater risk of additional weight gain [37,38], partially because of an increase in the prevalence of dysfunctional eating habits [24,39]. This worldwide phenomenon is so important that it was called “covibesity” [40]. Obesity is an important risk factors for several chronic diseases, including diabetes, cardiovascular disease, cancer, and also COVID-19 [41,42].

Only 34% of our sample agreed with WHO guidelines for PA, with a higher percentage (64%) among children at elementary school than older ones. Previous studies conducted in Canada and the U.S. found adolescents’ PA patterns to be more impacted by COVID-19 restrictions compared to younger children [43,44]. A possible explanation may be that younger children increased participation in outdoor independent activities, while high school-aged adolescents did not [44]. Our result confirmed the important role of sport in helping children, adolescents, and young adults maintain enough levels of PA, as found in a previous large European survey [45,46]. Increasing daily screen time was also associated with increasing inactivity, with probably a causal direct and reverse effect. The detrimental effect of high screen time on mental and physical health was previously reported [33] and represents a critical issue, especially in childhood and adolescence.

Our study was able to provide a complete picture on the impact of the COVID-19 pandemic on the health of young athletes, considering both the SARS-CoV-2 infection and parameters of good physical health. To the best of our knowledge, this is the first study putting together these aspects and it seems crucial because the decision of preventive measures should take into account the potential benefits, but also the associated risks. We also specifically addressed questions on preventive measures using a multivariable approach, which seems necessary to account for each specific measure independently from the others. The interesting result of a possible reduction of SARS-CoV-2 positivity in athletes participating in organized sport activity was so far never reported.

Among the limitations, the data were retrospectively collected through a self-administered online questionnaire, thus providing possible concerns about the data quality. We recognise this limitation, which was also discussed in other studies using similar surveys [45,47,48,49,50]. However, the high interpretability of our findings and consensus with other previous studies provides clues that the general quality was good. Waist circumference measurement in a clinical context may be a better indication of weight gain compared to BMI, since abdominal fat is associated with cardiovascular disease and metabolic syndrome. However, we used BMI in this study because this parameter is easier to measure by responders to surveys, without specific professional competence. Although other, not collected confounders could modify the results, we included important athletes’ and sports characteristics and a longer questionnaire could result in a lower compliance and accrual rate. Finally, the present results refer to the young Italian population and may not be necessarily extended to other geographical areas and age ranges.

## 5. Conclusions

In conclusion, we suggest that sport closure and limitations had an important negative impact on the overall health of young athletes, being also not effective in reducing the spread of COVID-19. Decision on preventive measures should be always based on scientific proof of efficacy and should always take into account the associated risks.

## Figures and Tables

**Figure 1 ijerph-19-07908-f001:**
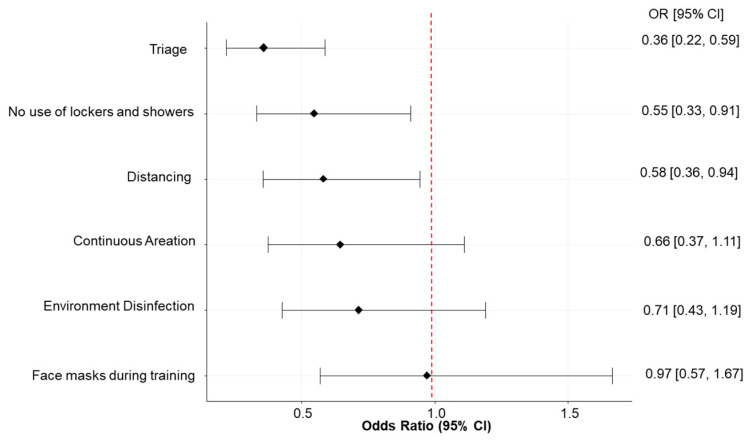
Odds ratio (OR) and 95% confidence intervals (CI) for the risk of being tested positive for SARS-CoV-2 according to the application of different preventive measures during sport activities. Note: triage consists in measuring fever and accounts for other symptoms and contact tracing.

**Figure 2 ijerph-19-07908-f002:**
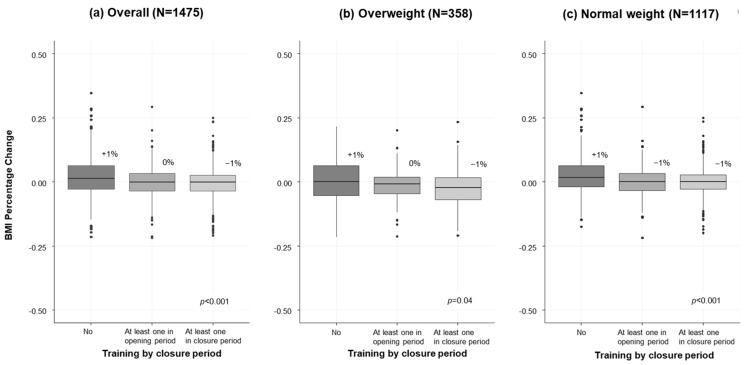
Box plots of BMI percent change of athletes according to participation in sport activities during opening and closing periods, for (**a**) all sample (N = 1475), and stratified by (**b**) overweight and (**c**) normal weight at the baseline.

**Table 1 ijerph-19-07908-t001:** Characteristics of the study population according to the absence or presence of at least one positive SARS-CoV-2 test in the whole study period.

Characteristics	All (N = 1750) ^$^	At Least One Positive SARS-CoV-2 Test (N = 192, 11%)	No Positive SARS-CoV-2 Test (N = 1558, 89%)^-^	*p*-Value *
**Age ^∞^**	12 (9, 15)	13 (10, 18)	12 (9, 15)	**<0.001**
**Gender**				0.34
Female	780 (45%)	78 (10%)	702 (90%)	
Male	944 (55%)	109 (12%)	835 (88%)	
**Education**				**<0.001**
Elementary school or lower	719 (41%)	53 (7%)	666 (93%)	
Middle school	427 (24%)	50 (12%)	377 (88%)	
High school	297 (17%)	23 (8%)	274 (92%)	
Professional high school	77 (4%)	14 (18%)	63 (82%)	
University	152 (9%)	28 (18%)	124 (82%)	
Other or no school	75 (4%)	21 (28%)	54 (72%)	
**Any training**				0.31
No	492 (28%)	59 (12%)	433 (88%)	
Yes	1258 (72%)	133 (11%)	1125 (89%)	
**Training by closure period ^**				0.55
No	492 (28%)	59 (12%)	433 (88%)	
At least one in opening period	305 (17%)	29 (10%)	276 (90%)	
At least one in closure period	953 (54%)	104 (11%)	849 (89%)	
**Weekly training sessions**				0.10
No	492 (28%)	59 (12%)	433 (88%)	
Up to two	853 (49%)	80 (9%)	773 (91%)	
More than two	405 (23%)	53 (13%)	352 (87%)	
**Baseline BMI ^∞^**	18.9 (16.7–21.3)	19.6 (17.6–22.0)	18.9 (16.6–21.3)	**<0.001**
**Geographical area**				0.65
North	1127 (66%)	113 (10%)	1014 (90%)	
Center	277 (16%)	26 (9%)	251 (91%)	
South and Islands	272 (16%)	33 (12%)	239 (88%)	
Abroad	30 (2%)	4 (13%)	26 (87%)	
**Number of SARS-CoV-2 tests ^∞^**	1 (0–3)	2 (1–5)	1 (0–2)	**<0.001**
**Outdoor physical activity**				0.05
No	549 (32%)	48 (9%)	501 (91%)	
Yes	1194 (67%)	143 (12%)	1051 (88%)	
**Highest parents’ education**				**<0.001**
Middle school or lower	50 (3%)	17 (34%)	33 (66%)	
High school	560 (33%)	67 (12%)	493 (88%)	
University or higher	1112 (64%)	103 (9%)	1009 (91%)	
**Parents’ workplace**				**0.03**
Home	345 (20%)	26 (8 %)	319 (92%)	
At least partially outside	1383 (80%)	161 (12%)	1222 (88%)	
**Mediterranean Diet**				**<0.001**
Adherence	523 (47%)	38 (7%)	485 (93%)	
No adherence	898 (63%)	121 (13%)	777 (87%)	

Abbreviations: BMI = body mass index. Note: significant *p*-values are in bold. Missing values are as follows: SARS-CoV-2 test (N = 14, excluded from the analysis), sex (N = 26), geographical area (N = 44), number of SARS-CoV-2 tests (N = 18), education (N = 3), highest parents’ education (N = 28), parents’ workplace (N = 22); and Column percentages; **^$^** row percentages; * chi-square or Fisher exact test, as appropriate, for the difference between subjects with no or any positive COVID-19 test; ^ opening periods: September/October 2020 and April/May 2021; closure period: November 2020 to March 2021; ^∞^ median (interquartile range).

**Table 2 ijerph-19-07908-t002:** Multivariable analysis for the risk of being tested positive for SARS-CoV-2 in the overall study population and in the subgroup of athletes enrolled in sports centers. Analysis comparing athletes with training sessions in opening and closing periods.

	Overall Analysis (*n* = 1732)	Enrolled in Sport Centers (*n* = 1669)
	**OR [95% CI]**	**OR [95% CI]**
**Training by closure period ^**		
No	1.00 [reference]	1.00 [reference]
At least one in opening period	1.00 [0.60, 1.65]	0.93 [0.56, 1.55]
At least one in closure period	0.89 [0.61, 1.30]	**0.62 [0.41, 0.93]**
**Gender**		
Female	1.00 [reference]	1.00 [reference]
Male	**1.47 [1.05, 2.07]**	1.42 [0.98, 2.04]
**Number of SARS-CoV-2 tests ***	**1.16 [1.11, 1.21]**	**1.19 [1.13, 1.24]**
**Education**		
Elementary school or lower	1.00 [reference]	1.00 [reference]
Middle school	1.22 [0.80, 1.88]	1.19 [0.76, 1.87]
High school	0.79 [0.46, 1.36]	0.83 [0.47, 1.44]
Professional high school	**2.16 [1.09, 4.27]**	1.95 [0.94, 4.06]
University	**2.30 [1.32, 4.01]**	1.57 [0.85, 2.89]
Other or no school	**3.31 [1.75, 6.28]**	1.81 [0.86, 3.81]
**Highest parents’ education**		
University or higher	1.00 [reference]	1.00 [reference]
High school	1.22 [0.86 1.73]	1.21 [0.83, 1.75]
Middle school or lower	**3.63 [1.82, 7.27]**	**3.85 [1.81, 7.97]**
**Parents’ workplace**		
Home	1.00 [reference]	1.00 [reference]
At least partially outside	**1.57 [1.00, 2.42]**	**1.63 [1.00, 2.67]**

Abbreviations: CI = confidence interval; OR = odds ratio; note: significant odds ratios are in bold; ^ opening periods: September/October 2020 and April/May 2021; closure period: November 2020 to March 2021; * N = 18 missing. Reported odds ratio by unit increase.

**Table 3 ijerph-19-07908-t003:** Multivariable analysis for the risk of at least one person among cohabitants tested positive for SARS-CoV-2 in the overall study population and in the subgroup of athletes enrolled in sports centers. Analysis comparing families of athletes with training sessions in opening and closing periods.

	Overall Analysis (*n* = 1730)		Enrolled in Sport Centers (*n* = 1668)	
	N (% at Least One Positive Test among Cohabitants)	OR [95% CI]	N (% at Least One Positive Test among Cohabitants)	OR [95% CI]
**Training by closure period ^**				
No	488 (18%)	1.00 [reference]	488 (18%)	1.00 [reference]
At least one in opening period	304 (18%)	1.20 [0.80, 1.80]	284 (18%)	1.09 [0.72, 1.64]
At least one in closure period	938 (19%)	0.98 [0.71, 1.35]	896 (17%)	0.78 [0.56, 1.08]
**Gender**				
Female	772 (17%)	1.00 [reference]	741 (16%)	1.00 [reference]
Male	934 (19%)	1.22 [0.93, 1.59]	903 (18%)	1.20 [0.90, 1.60]
**Geographical area**				
North	1116 (18%)	1.00 [reference]	1082 (17%)	1.00 [reference]
Center	273 (14%)	**0.61 [0.40, 0.90]**	267 (14%)	**0.64 [0.42, 0.96]**
South and Islands	270 (19%)	1.13 [0.79, 1.62]	255 (17%)	1.00 [0.68, 1.48]
Abroad	29 (24%)	1.01 [0.37, 2.75]	27 (26%)	1.19 [0.44, 3.28]
**Number of SARS-CoV-2 tests ***	-	**1.20 [1.15, 1.25]**	-	**1.21 [1.16, 1.27]**
**Education**				
Elementary school or lower	717 (14%)	1.00 [reference]	691 (13%)	1.00 [reference]
Middle school	424 (21%)	1.25 [0.89, 1.74]	410 (21%)	1.28 [0.91 1.81]
High school	289 (12%)	0.62 [0.40, 0.96]	284 (12%)	0.64 [0.41, 1.00]
Professional high school	78 (29%)	**1.82 [1.03, 3.21]**	75 (28%)	**1.84 [1.02, 3.33]**
University	147 (24%)	**1.64 [1.01, 2.66]**	141 (21%)	1.29 [0.77, 2.18]
Other or no school	72 (40%)	**3.09 [1.74, 5.49]**	65 (34%)	**2.16 [1.15, 4.07]**
**Number of cohabitants**	-	**1.17 [1.02, 1.33]**	-	**1.16 [1.01, 1.33]**
**Highest parents’ education**				
University or higher	1102 (17%)	1.00 [reference]	1067 (16%)	1.00 [reference]
High school	555 (20%)	1.14 [0.86, 1.51]	533 (20%)	1.11 [0.83, 1.50]
Middle school or lower	48 (38%)	**2.29 [1.17, 4.45]**	45 (18%)	**2.25 [1.13, 4.50]**
**Parents’ workplace**				
Home	339 (14%)	1.00 [reference]	328 (12%)	1.00 [reference]
At least partially outside	1369 (19%)	**1.60 [1.11, 2.30]**	1318 (18%)	**1.68 [1.14, 2.46]**

Abbreviations: CI = confidence interval; OR = odds ratio; note: significant odds ratios are in bold; ^ opening periods: September/October 2020 and April/May 2021; closure period: November 2020 to March 2021; * N = 18 missing. Reported odds ratio by unit increase.

## Data Availability

The datasets used and/or analysed during the current study are available from the corresponding author on reasonable request.

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
