# Peer review of "The Impact of Sport Activity Shut down during the COVID-19 Pandemic on Children, Adolescents, and Young Adults: Was It Worthwhile?"

_ijerph, 2022, doi:10.3390/ijerph19137908_

Round 1

Reviewer 1 Report

It is an important paper on the important topic of SARS-CoV-2-related decision of sports shut down as a preventive measure. The paper is elegantly drafted and presents interesting data, underlying that sports play an important role in the physical and mental well-being of children and adolescents. However, I can not find on sixteen pages of the Supplementary Materials, the original version of the questionnaire. Maybe it's my mistake. Farther, I only have a couple of minor suggestions to improve the readability of the manuscript:

L 30 All abbreviations should be expended when used for the first time. Even BMI and WHO.

L 38-39 As "Thus, we strongly recommend avoiding sport closure and limitations for children, adolescent and young adults." is not directly related with the aim of the study. It would be better to rewrite or remove this sentence.

In the material and methods section, I recommend moving some detailed data of the respondents from the supplementary materials to the main section. It would be advisable that the material and methods section should not consist only of a brief introduction and description of the statistical analysis.

 As in line 102 change "we calculated Odds Ratio (OR) with" to the passive voice throughout the manuscript

L 123-127 This paragraph should be moved to the material and method section.

I agree with the authors' statement that this is the first national Italian study that investigated the impact of sport closure on SARS-239 CoV-2 infection and the overall health of young athletes. Therefore, the limited discussion section is justified.

L 320-321 Similar to the introduction section, "Based on our results, we strongly recommend avoiding sport closure and limitations for children, adolescent and young adults, keeping effective preventive safety measures" is not directly related to the aim of the study. It would be better to rewrite or remove this sentence.

Summarizing the overall reception of the manuscript, it is well done!

Author Response

It is an important paper on the important topic of SARS-CoV-2-related decision of sports shut down as a preventive measure. The paper is elegantly drafted and presents interesting data, underlying that sports play an important role in the physical and mental well-being of children and adolescents.

ANSWER: We would like to thank the reviewer for the positive comments on our manuscript.

However, I can not find on sixteen pages of the Supplementary Materials, the original version of the questionnaire. Maybe it's my mistake.

ANSWER: The questionnaire was uploaded as a separate file in the 7-Zip compressed folder.

Farther, I only have a couple of minor suggestions to improve the readability of the manuscript:

L 30 All abbreviations should be expended when used for the first time. Even BMI and WHO.

ANSWER: We have expanded the abbreviations throughout the revised paper and in the abstract.

L 38-39 As "Thus, we strongly recommend avoiding sport closure and limitations for children, adolescent and young adults." is not directly related with the aim of the study. It would be better to rewrite or remove this sentence.

ANSWER: We removed the sentence in the abstract and in the conclusions, as suggested.

In the material and methods section, I recommend moving some detailed data of the respondents from the supplementary materials to the main section. It would be advisable that the material and methods section should not consist only of a brief introduction and description of the statistical analysis.

ANSWER: We thank the reviewer for this suggestion. We have moved some more details on the questionnaire to the materials and methods in the main text.

 As in line 102 change "we calculated Odds Ratio (OR) with" to the passive voice throughout the manuscript

ANSWER: We change the active to the passive form, as suggested, in a couple of sentences in the materials and methods section 

L 123-127 This paragraph should be moved to the material and method section.

ANSWER: We moved the paragraph to the material and method section, as suggested.

I agree with the authors' statement that this is the first national Italian study that investigated the impact of sport closure on SARS-239 CoV-2 infection and the overall health of young athletes. Therefore, the limited discussion section is justified.

ANSWER: Thanks for the comment.

L 320-321 Similar to the introduction section, "Based on our results, we strongly recommend avoiding sport closure and limitations for children, adolescent and young adults, keeping effective preventive safety measures" is not directly related to the aim of the study. It would be better to rewrite or remove this sentence.

ANSWER: We removed the sentence in the abstract and in the conclusions, as suggested.

Summarizing the overall reception of the manuscript, it is well done!

ANSWER: Thank you very much for the kind and encouraging positive comments on our manuscript!

Reviewer 2 Report

Methods

Line 80- Body mass index was used to assess weight gain of individuals, but can the authors explain why this mode of measurement was used. Waist circumference measurement may have been a better indication of weight gain since abdominal fat is associated with cardiovascular disease and metabolic syndrome.

Line 80- What parameters were used to determine a positive or negative COVID test result (i.e., did the COVID test have to be performed in the doctor’s office or were at-home COVID tests also used?)

Line 80- How were false positive and negative COVID test results controlled for?

Line 80- If an individual tested positive for COVID, was the severity of COVID symptoms tracked? In addition, was the reoccurrence of COVID tracked in individuals who tested positive?

Line 80- Was COVID vaccination status among individuals accounted for?

Author Response

Methods

Line 80- Body mass index was used to assess weight gain of individuals, but can the authors explain why this mode of measurement was used. Waist circumference measurement may have been a better indication of weight gain since abdominal fat is associated with cardiovascular disease and metabolic syndrome.

ANSWER: We agree with the reviewer that waist circumference measurement in a clinical context may have been a better indication of weight gain compared to BMI. However, we need a parameter that could be easily measured by individuals without specific professional competence and reported in the survey. We have added this observation in the discussion section of the revised version of the manuscript (lines 348-352).

Line 80- What parameters were used to determine a positive or negative COVID test result (i.e., did the COVID test have to be performed in the doctor’s office or were at-home COVID tests also used?)

ANSWER: In Italy, in the study period (Oct 2020-May 2021), at-home COVID tests could not be used to determine COVID test results, therefore the reported positive tests referred only to results obtained by general practitioner or pharmacy or territorial swab point or in health facilities.

Line 80- How were false positive and negative COVID test results controlled for?

ANSWER: In Italy, cases where the antigenic test result was not confirmed by the molecular test within 48 hours were considered as false positives or false negatives.

Line 80- If an individual tested positive for COVID, was the severity of COVID symptoms tracked? In addition, was the reoccurrence of COVID tracked in individuals who tested positive?

ANSWER: Since data of this study were collected via a web, self-administered, survey, we avoided collecting too many details in order to not discourage athletes to participate. Indeed, during the two-weeks pilot phase, we received feedback by most of the adolescents and young adults, who suggested shortening the questionnaire to increase the compliance. We therefore decided to focus on the main variables related to the outcomes of Covid-19 infection and athletes well-being.

Line 80- Was COVID vaccination status among individuals accounted for?

ANSWER: We would like to thank the reviewer for this comment. In Italy, in the study period (Oct 2020-May 2021), the percentage of vaccinated young athletes was negligible. Indeed, Covid-19 vaccination for children under 11 years of age was available much later (Dec 2021). For over-12 athletes, the vaccination campaign in Italy started only in June 2021, unless for frail persons, but they reasonably represents a very small fraction of our population of healthy athletes.

In recognition of the importance of this observation, we have added a sentence in the discussion section to highlight and clarify this point (lines 286-290).